# Characterization of Cross-Linked Enzyme Aggregates of the Y509E Mutant of a Glycoside Hydrolase Family 52 β-xylosidase from *G. stearothermophilus*

**DOI:** 10.3390/molecules26020451

**Published:** 2021-01-16

**Authors:** Gabriela Romero, Lellys M. Contreras, Carolina Aguirre, Jeff Wilkesman, Josefa María Clemente-Jiménez, Felipe Rodríguez-Vico, Francisco Javier Las Heras-Vázquez

**Affiliations:** 1Center for Environmental, Biological and Chemical Research, Experimental Faculty of Sciences and Technology, University of Carabobo, Valencia 2001, Venezuela; gaby32004@gmail.com (G.R.); mariela@ual.es (L.M.C.); wilkesman@lba.hs-mannheim.de (J.W.); 2Department of Chemistry and Physics, University of Almeria, Building CITE I, Carretera de Sacramento s/n, La Cañada de San Urbano, 04120 Almería, Spain; jmclemen@ual.es (J.M.C.-J.); fvico@ual.es (F.R.-V.); 3Centro de Investigación en Biodiversidad y Ambientes Sustentables (CIBAS), Department of Environmental Chemistry, Faculty of Sciences, Universidad Católica de la Santísima Concepción, Casilla 297, Concepción 4090541, Chile; caguirre@ucsc.cl; 4Institute for Biochemistry, University of Applied Sciences Mannheim, Paul-Wittsack-Straße 10, D-68163 Mannheim, Germany; 5Campus de Excelencia Internacional Agroalimentario ceiA3, University of Almeria, 04120 Almería, Spain

**Keywords:** β-xylosidase, thermostability, CLEAs, *G. stearothermophilus*, xylanase

## Abstract

Cross-linked enzyme aggregates (CLEAs) of the Y509E mutant of glycoside hydrolase family 52 β-xylosidase from *Geobacillus stearothermophilus* with dual activity of β-xylosidase and xylanase (XynB2^Y509E^) were prepared. Ammonium sulfate was used as the precipitant agent, and glutaraldehyde as cross-linking agent. The optimum conditions were found to be 90% ammonium sulfate, 12.5 mM glutaraldehyde, 3 h of cross-linking reaction at 25 °C, and pH 8.5. Under these (most effective) conditions, XynB2^Y509E^-CLEAs retained 92.3% of their original β-xylosidase activity. Biochemical characterization of both crude and immobilized enzymes demonstrated that the maximum pH and temperature after immobilization remained unchanged (pH 6.5 and 65 °C). Moreover, an improvement in pH stability and thermostability was also found after immobilization. Analysis of kinetic parameters shows that the *K*_m_ value of XynB2^Y509E^-CLEAs obtained was slightly higher than that of free XynB2^Y509E^ (1.2 versus 0.9 mM). Interestingly, the xylanase activity developed by the mutation was also conserved after the immobilization process.

## 1. Introduction

Enzyme immobilization offers the prospect of reusability and enhances the enzyme stability [1]. Immobilization can be conducted binding to a solid-carrier, encapsulation in a matrix, or by cross-linking enzyme molecules [2]. Currently, the preparation of cross-linked enzymatic aggregates (CLEAs) has gained relevance as a strategy for enzyme immobilization [1,2]. The CLEAs methodology is an alternative to traditional immobilization methods on pre-existing solid supports that could reduce the specificity and activity of the biocatalyst [2]. This method is also attractive because it is simple and robust, and the prepared CLEAs exhibit improved operational stability and excellent catalytic activity [3]. 

Fundamentally, the immobilization of the enzymes as CLEAs involves two steps: Precipitation of the soluble enzyme and cross-linking of the formed aggregates. In the first step, the semi-purified soluble enzyme is precipitated (without perturbation of the tertiary structure) by addition of precipitating agents such as salts or organic solvents miscible with water [4,5]. Then, the formed aggregates are chemically cross-linked using a bifunctional reagent that allows reactions of amino groups of exposed lysine residues on the enzyme surface. After chemical cross-linking, it is important that the cross-linked aggregate remains insoluble, maintaining its structure, as well as its catalytic activity. 

The CLEAs procedure is applicable to essentially any enzyme. The greater number of CLEAs described in the literature are related to hydrolases, basically because these enzymes have a variety of industrial applications [6]. Among them, the successfully immobilization of glycosidases in CLEAs has been reported [7,8,9,10]. The union of CLEA technology with multipurpose and cascade biocatalysis has proven to be particularly attractive for practical applications [1,6]. For example, a novel combi-CLEAs comprising the enzymes xylanase, cellulase, and β-1,3-glucanase was used for one-pot saccharification of sugarcane bagasse [11]. The combi-CLEAs strategy has also been applied in cofactor recycling. A prominent example is the glycerol dehydrogenase (GDH) and NADH oxidase (Nox) combi-CLEAs that was successfully used as a NAD^+^ regeneration system for the synthesis of 1,3-dihydroxyacetone (DHA) [12]. 

From *G. stearothermophilus,* a β-xylosidase belonging to the CAZy glycoside hydrolase family GH52 (XynB2) has been biochemically and biophysically well characterized [13,14,15,16,17]. Different residues of this enzyme have been replaced via site-directed mutagenesis in order to know the catalytic nucleophile and acid-base residues of family 52 glycoside hydrolases [15] and its glycosynthase activity [18]. Recently, Huang et al. identified that the tyrosine 509 (Y509) site of XynB2 overlapped with the glutamic acid 506 (E506) site of xylA and the glutamic acid 298 (E298) site of xylB. The mutation Y509E resulted in an enzyme variant that showed both xylosidase and xylanase activity [19]. These reports highlighted that XynB2 could be a valuable enzyme for industrial purposes, especially in biorefineries and the food industry for bioconversion of lignocellulosic biomass into ethanol and xylitol.

The CLEA technology has been defined as an appealing candidate for the bioconversion of lignocellulosic biomass. For example, CLEAs of β-mannanase prepared using linear macromolecular cross-linkers instead of micro-molecular glutaraldehyde resulted in a porous structure with low steric hindrance and higher activity [20]. Bhattacharya and Pletschke reported the use of magnetite-CLEAs in the field of lignocellulose bioconversion. In comparison with the conventional CLEAs, the magnetite-CLEAs displayed higher xylanase activity [21]. Lately, the same group prepared a combi-CLEAs comprising of xylanases and mannanases produced by native bacterial strains [22]. The thermal stability of all enzymes into CLEAs was increased compared to the free enzymes. Moreover, the concentration of sugars released following the hydrolysis of lime pre-treated bagasse and corn stover by xylanase-mannanase combi-CLEAs was higher in comparison with the release of monomeric sugar by individual CLEAs. More recently, a *Conhella* sp. AR92 xylanase CLEA (Xyl-CLEA) was applied successfully to the hydrolysis of pre-treated sugarcane bagasse [9]. Interestingly, the Xyl-CLEAs synthesized displayed a different specificity compared to the free enzyme. 

In the present work, a β-xyloxidase from *Geobacillus stearothermophilus* was engineered by site-directed mutagenesis to introduce a new exo-xylanase activity. XynB2^Y509E^ mutant was produced and its activity as a xylanase and as β-xylosidase was tested. CLEAs were prepared in order to evaluate the effect of the immobilization as CLEAs on the β-xylosidase ability of the mutant Y509E. Different parameters such as the type of precipitant, precipitation time, type and concentration of cross-linker, and crosslinking reaction time were optimized. The biochemical properties of XynB2^Y509E^-CLEAs were assessed and compared to those of the corresponding soluble enzyme by determining the pH and temperature values with maximal activity, thermostability, pH stability, kinetic parameters, and reusability. To the best of our knowledge, this is the first report of immobilization of a thermostable β-xylosidase as cross-linked enzyme aggregates. 

## 2. Results and Discussion

### 2.1. Optimization of XynB2^Y509E^ CLEAs Preparation

#### 2.1.1. Optimization of Precipitant Type and Precipitation Time

CLEAs synthesis procedure begins with the formation and precipitation of protein aggregates, caused by various precipitating agents such as organic solvents, polymers, and salts. The formation of these aggregates occurs by changes in hydration states of molecules or alteration of electrostatic forces in the solution. Then, aggregates are transformed into a more stable structure using a cross-linking agent, which establishes covalent bonds between protein molecules, which makes it insoluble [2]. In this study, butanol, acetone, and ammonium sulfate were used for the precipitation of XynB2^Y509E^ (Figure 1a). To select the suitable precipitant agent, we considered the yield of good aggregates and its recovery, as well as the preservation of catalytic activity. Although the precipitation with butanol was effective, the precipitate remains in suspension which prevents its clean and complete recovery. The precipitate obtained with acetone was cleaner and easier to recover than that obtained with butanol, though the enzyme completely lost catalytic activity. Moreover, the best result (recovery of more than 90% of the initial β-xyloxidase activity and homogeneous precipitation that permit the easy recovery of the aggregates) was obtained using ammonium sulfate as precipitant. To ensure a more complete protein precipitation, the final saturation of ammonium sulfate was increased to 90%. This result is consistent with a recent study that suggests that the use of high precipitant concentrations avoids the risk of partial inactivation of protein due to incomplete aggregation [10,23].

Since the precipitation time-step plays a fundamental role in activity recovery [4], different aggregation times were also evaluated. Accordingly, the aggregation process was carried out for 30, 60, and 120 min. We settled the efficiency of these aggregation times by measuring the amount of protein in the supernatant and assaying its β-xylosidase activity. The results showed that the increase in precipitating time resulted in a slight increase in the activity recovery (Figure 1b). The time required for precipitation varies among different proteins. However, it is thought that when the precipitation process is slow, the enzyme can become denatured, while if it happens quickly, there is a greater chance that the enzyme molecule will be able to find neighboring molecules in time and preserve its tertiary structure [4]. 

The enzyme purity is another critical parameter to be considered during the immobilization process. As CLEAs methodology combines purification and immobilization into one step, the enzyme does not need to be of high purity [2]. Therefore, in this study, we used a semi-purified enzyme, i.e., an enzyme preparation obtained after the heat-treatment. Recently, the production of a CLEA consisting of cellulase and xylanase, which was prepared from crude enzyme solutions, has been reported [24]. Interestingly, this paper reported an enhanced enzyme activity of immobilized crude enzymes in comparison with the CLEAs produced from high purity commercial enzymes, suggesting that the approach using crude or semi-purified enzyme could be an advantage for the saccharification process. 

#### 2.1.2. Effect of Concentration and pH in Cross-Linking Reaction

Glutaraldehyde is one of the most frequently used cross-linker in CLEAs preparation. Given that some enzymes are inactivated by glutaraldehyde, the cross-linker amount and cross-linking time on the activity of the resultant CLEAs must be optimized. In the present study, different concentrations of glutaraldehyde (12.5, 25, and 40 mM) were employed for 3 h. Figure 1c shows the relative activity in the CLEAs. The three different glutaraldehyde concentrations tested showed the maximum of the relative activity expected. Thereby, 12.5 mM was selected as the optimal cross-linker concentration of glutaraldehyde. Subsequently, the optimal cross-linking time was investigated within the range from 2 to 4 h. As shown in Figure 1d, the relative activity increased with the prolonging of cross-linking time. Thus, 100% of relative activity was observed at 3 h. Based on this result, 3 h of cross-linking time was used in all further experiments.

It is known that the pH has an effect on the polymerization of glutaraldehyde molecules in the solution [9]. The preparation of CLEAs is commonly carried out at a pH range from nearly neutral up to slightly alkaline [4]. This choice is due to the high reactivity of glutaraldehyde toward proteins at neutral pH. Furthermore, glutaraldehyde structure in aqueous solution at acidic pH is predominantly in its monomeric form, whereas alkaline pH promotes the conversion to the polymeric form [25]. In addition, it was previously demonstrated that the pH value of glutaraldehyde solution had remarkable effects on the particle size of lipases CLEAs [26]. Indeed, the CLEAs prepared with glutaraldehyde at pH 4.5 are compact (~5 µm) with less space between enzyme aggregates, whereas that the CLEAs produced with glutaraldehyde at pH 9.5 are bigger (~20 µm) with more space between the enzyme aggregates. 

Regarding the XynB2^Y509E^ mutant under study, it is important to emphasize that in its free form, it shows both xylosidase and xylanase activity. The xylanase activity displayed by the XynB2^Y509E^ mutant was firstly demonstrated by Huang et al., where it was hypothesized that the glutamic acid substitution might introduce some flexibility into the orientation of the xylan chain in the substrate binding cleft, but the mutation might also widen the active site pocket [19].

The theoretical p*I* of this mutant enzyme was determined to be 5.19, with a protein charge of −20 at pH 6.5 and −40 at pH 8.5. According to this and other considerations mentioned above, the pH effect of the glutaraldehyde solution during the cross-linking reaction was also studied. Figure 2 is the scanning electron micrograph (SEM) of the XynB2^Y509E^-CLEAs prepared at two different pH values. By SEM observation, the CLEAs prepared at pH 6.5 (Figure 2a) showed a smoother surface whereas the CLEAs at pH 8.5 (Figure 2b) appeared as a closer-packed structure exposing more cavities. In both conditions, the enzyme molecules were packed together into spheres and are similar to that of the type 1 aggregates described by Schoevaart et al. [23]. At pH 6.5, the size of the enzyme microaggregates was about 1 µm, whereas at pH 8.5, it was 75% bigger (1.75 µm). These results suggest that the pH value seems to have a relevant effect on the morphology and size of the microaggregates. Activity assays demonstrated that the β-xylosidase activity was retained in the CLEAs prepared with glutaraldehyde at pH 6.5 or pH 8.5. However, the xylanase activity was only preserved in the CLEAs prepared at pH 8.5. The xylanase activity lost in the CLEAs prepared with glutaraldehyde at pH 6.5 can be attributed to the diffusional problem of xylan with the supramolecular structure [27]. In agreement with this, the diffusional problems could minimize if the spatial structure of CLEAs is enlarged. Thus, we suggest that bigger size pore in the CLEAs produced with glutaraldehyde at pH 8.5 prevents the occurrence of diffusional restrictions. Thereby, the preservation of xylanase activity is possibly due to an increase in the pore diameter of the CLEAs prepared at pH 8.5. 

In summary, the optimal conditions for the XynB2^Y509E^-CLEAs preparation were protein precipitation with 90% ammonium sulfate for 30 min and crosslinking using 12.5 mM glutaraldehyde at pH 8.5 during 3 h. In another way, we also evaluated the protein recovery to characterize the global yield of the cross-linking process [12]. The aggregation yield value determined for the XynB2^Y509E^-CLEAs prepared under those optimal conditions was found in 87.3%. The high yield obtained allows us to conclude that the cross-linking process carried out was effective. Furthermore, this result is completely consistent with the high value of recovered activity of the XynB2^Y509E^-CLEAs.

### 2.2. Biochemical Characterization of Free XynB2^Y509E^ and XynB2^Y509E^-CLEAs

#### 2.2.1. Effect of Reaction Times on both Xylanolytic Activities 

In order to determine an optimal reaction time of the free and immobilized enzyme, the two enzyme activities were assayed by varying the time of enzyme-substrate incubation. For the xylanase activity, the reaction time ranged from 30 to 180 min (Figure 3a). As mentioned earlier, XynB2^Y509E^-CLEAs prepared at pH 6.5 were not able to degrade xylan either after 30 min, nor 60, 120, and 180 min of reaction time. The xylanase activity of the XynB2^Y509E^-CLEAs prepared at pH 8.5 showed optimum reaction time of 180 min, which is longer than the free XynB2^Y509E^ of 120 min. This increase in the reaction time could be due to the fact that the xylan requires more time to penetrate into the CLEAs in order to access the active site. Interestingly, it has been also reported that the reaction time of a xylanase was increased from 5 to 60 min when it was immobilized within calcium alginate beads using an entrapment technique [28]. Figure 3b shows the effect on the reaction time of the β-xylosidase activity for the free and immobilized enzymes. The free and immobilized enzymes both exhibited a maximum activity at 5 min of reaction time. Due to the fact that XynB2^Y509E^-CLEAs obtained at pH 8.5 did not recover 100% of its xylanase activity, not even after 180 min of reaction time, the biochemical characterization proceeded by measuring only the β-xylosidase activity.

#### 2.2.2. Effect of the Temperature and pH on Activity and Stability 

The effect of temperature on the β-xylosidase activity of free XynB2^Y509E^ and XynB2^Y509E^-CLEAs is shown in Figure 4a. The maximal temperature of the immobilized XynB2^Y509E^ was 65 °C, the same as that of the free enzyme. However, the immobilization preserved greater activities at higher temperatures (>70 °C) than that of the equivalent free counterpart. For instance, at 75 °C, the free enzyme retained 85% of its relative activity, while the XynB2^Y509E^-CLEAs presented 97% of its initial activity. Furthermore, the XynB2^Y509E^-CLEAs was found to preserve 90% of its initial activity at 80 °C, whereas the free enzyme showed only 44% relative activity at the same temperature. Similar results have been found by Verma et al., who reported a larger resistance to higher temperatures for CLEAs xylanase in comparison to free xylanases [10]. 

The thermo stability of the XynB2^Y509E^ and XynB2^Y509E^-CLEAs was investigated by incubating the enzymes in the absence of substrate for 1 h between 30 °C and 90 °C. As shown in Figure 4b, XynB2^Y509E^-CLEAs revealed a compelling increase of thermal stability in comparison to free XynB2^Y509E^. XynB2^Y509E^-CLEAs were stable up to 70 °C, whereas the free enzyme lost 22% at 50 °C and more than 80% at 70 °C. In addition, after the incubation for 1 h at 75 °C, XynB2^Y509E^-CLEAs retained more than 75% of its activity. The improved thermal stability presented by CLEAs was reported earlier. For example, the CLEAs of poly-3-hydroxybutyrate depolymerase retained 50% activity after incubation at 70 °C for 40 min, while the free enzyme was completely deactivated after incubation under the same condition [29]; the CLEAs of laccases from two fungal strains retained more than 50% of their activity at 50 °C, whereas free laccases rapidly decreased in activity after 40 °C [30]; the CLEAs of xylanase from *Geobacillus thermodenitrificans* X1 maintained 53% of activity after 4 h at 70 °C in comparison to free xylanase that retained only 15% of activity [10]. This enhancement of thermal stability CLEAs may be partially attributed to the covalent cross-linking among enzyme aggregates. 

The effect of pH on the β-xylosidase activity profiles of XynB2^Y509E^ and XynB2^Y509E^-CLEAs were assessed in a pH range from 4.0 to 11.0, and results are displayed in Figure 5a. Although both enzymes showed an optimal pH of 6.5, at acidic pH (4.5 and 5) or at pH between 7 and 9.5, the CLEAs exhibited a higher activity in comparison to the free form. In order to get more details on this effect, the XynB2^Y509E^ and XynB2^Y509E^-CLEAs stability were studied during their storage at 4 °C for 1 h at different pH values. As indicated in Figure 5b, XynB2^Y509E^-CLEAs upheld a stability that was significantly higher than the free enzyme. At pH 8.0, XynB2^Y509E^-CLEAs retained 92% activity in comparison to free XynB2^Y509E^ that retained 70% of activity. In addition, the immobilized biocatalyst retained over 60% of activity at pH 9.0, whereas the free enzyme preparation retained 45% of its activity in identical conditions. Moreover, XynB2^Y509E^-CLEAs maintained approximately 48% activity when stored at pH 9.5, while its free counterpart only maintained 35% of its activity. The resistance to alkaline pHs has also been demonstrated for other CLEAs [31,32]. These authors explained this resistance as a consequence of the crosslinking process, where most of the available amino groups on the surface of the enzyme engaged with glutaraldehyde, and hence a higher pH might be beneficial for stabilizing the enzyme by neutralizing the acidic groups on the enzyme surface. 

From the temperature and pH stability tests performed, it may be concluded that the stability of β-xylosidase could be improved through the immobilization with the method of cross-linked enzyme aggregates.

#### 2.2.3. Kinetic Parameters for the Free XynB2^Y509E^ and XynB2^Y509E^-CLEAs

Kinetic parameters for β-xylosidase activity of free XynB2^Y509E^ and XynB2^Y509E^-CLEAs were determined by calculating initial rates at various p-NPX concentrations. It was observed that the free enzyme and the immobilized enzyme both follow the Michaelis–Menten kinetics model. Table 1 shows K_m_ and V_max_ values for β-xylosidase activity of XynB2^Y509E^ free and immobilized in CLEAs using p-NPX as substrate. The enzyme immobilized in CLEAs exhibited a slightly higher increase in K_m_ value than that of the free enzyme (1.2 ± 0.2 mM vs. 0.9 ± 0.1 mM), which could be attributed to the reduced accessibility of the substrate caused by the crosslinking, as reported for other enzymes also immobilized by the same approach [7,8,33]. In fact, this finding is in correspondence with the increase in the reaction times observed for the xylanase activity of the immobilized mutant enzyme. On the other hand, the V_max_ value for the immobilized enzyme was 1.18 µmol/min, 25% lower than that of the free enzyme (1.6 µmol/min) as can be seen in Table 1. These results, higher K_m_ value with lower V_max_ value, have also been reported in other studies for immobilized enzyme in CLEAs [31,34]. Finally, from the kinetic study, it can be concluded that enzyme kinetic properties changed during the immobilization process.

#### 2.2.4. Reusability

One of the major advantages of the immobilized enzyme is reusability, which can reduce the amount of free enzyme in industrial production resulting in lower production costs. The reusability of XynB2^Y509E^-CLEAs was evaluated by consecutive cycles of assay. As shown in Figure 6, the immobilized enzyme retained 80% of their initial activities after 5 cycles and 60% at the tenth cycle of reuse. The decreased activity with increasing reuse cycle could be attributed to mechanical reasons that inevitably occur during washing and centrifugation [8,10,34,35]. These results suggest that XynB2^Y509E^-CLEAs have a good operational stability. However, it is necessary to evaluate its activity on other substrates (i.e., xylan) in order to have a wider view of its utility in industrial processes.

## 3. Materials and Methods

### 3.1. Materials

Ammonium sulfate and glutaraldehyde (50%) were obtained from Merck (Darmstadt, Germany) and Scharlab (Barcelona, Spain). All chemicals were of the highest purity available and of analytical grade. 

### 3.2. Mutagenesis of the xynB2 Gene

Site-directed mutagenesis was performed using the QuikChange II Site-directed mutagenesis kit (Stratagene, La Jolla, CA), following the manufacturer’s protocol, and using the plasmid pJAVI91 as a template [16]. The mutagenic primers for the Y509E mutation were as follows (the mutated nucleotides are in bold type): 5-GCGCGCAACAATTTA**GAG**TTGACAGGAAAAT-3′5-ATTTTCCTGTCAA**CTC**TAAATTGTTGCGCGC-3′

Mutation was confirmed by using the dye dideoxy nucleotide sequencing method in an ABI 377 DNA Sequencer (Applied Biosytems, Foster City, CA, USA). The plasmid containing the mutation Y509E was named pJAVI100. 

### 3.3. Overexpression and Partial Purification of XynB2^Y509E^

*E. coli* C43 was transformed with pJAVI100 plasmid. The cells carrying recombinant plasmid were cultured in LB broth supplemented with 100 µg/ml ampicillin at 37 °C, 250 rpm. After OD_600_ reached 0.5, IPTG was added to the final concentration of 0.1 mM and cells were further incubated at 37 °C, 250 rpm for 18 h. The cells were harvested by centrifugation then lysed by sonication. The cell extract was centrifuged at 10,000 rpm (Beckman Coulter J2-21, rotor JA20, Brea, CA, USA) for 30 min, and the soluble fraction was then heat-treated (45 °C, 30 min) and centrifuged again. The supernatant was dialyzed against 0.1 M citrate phosphate-glycine buffer (CFG), pH 6.5 (29.41 g C_6_H_5_O_7_Na_2_.2H_2_O, 13.80 g NaH_2_PO_4_ and 7.51 g NH_2_CH_2_COOH in 1 L distilled water) at 4 °C overnight.

### 3.4. Enzymatic Assays

β-xylosidase activity of free and XynB2^Y509E^-CLEAs was determined by assaying the amount of p-nitrophenol (pNP) released from the substrate p-nitrophenyl-β-d-xylopyranoside (pNPX). A reaction mixture of 5 µL of enzyme, 45 µL CFG buffer, and 50 µL of pNPX (2.2 mM) was incubated at 50 °C for 5 min. The reaction was stopped by addition of 100 µL of 1 M Na_2_CO_3_. The absorbance of the liberated p-nitrophenol was measured at 410 nm, by using the extinction coefficient Δε = 18 mM^−1^ cm^−1^. One unit of enzyme was defined as the amount of enzyme required to produce 1 µmol of p-nitrophenol per min under standard assay condition. All measurements were analyzed in triplicate. Xylanase activity was determined by measuring the amount of reducing sugars released from birchwood xylan employing the 3,5-dinitrosalicylic acid (DNS) assay according to Miller [36]. The standard reaction mixture contained 50 µL of xylan (1% *w*/*v* in 0.1 M CFG buffer, pH 6.5) and 50 µL of free and XynB2^Y509E^-CLEAs. Hydrolysis was conducted at 50 °C for 30 min for the free enzyme. However, for XynB2^Y509E^-CLEAs, the enzyme-substrate incubation time was increased to 1 h. The reaction was stopped by adding 100 µL of DNS reagent, and tubes were incubated at 100 °C for 15 min. Subsequently, 400 µL of distilled water was added in the reaction tubes and mixed thoroughly. The absorbance of the liberated xylose was measured at 540 nm, by using the extinction coefficient Δε = 0.082 mM^−1^ cm^−1^. One unit of enzymatic activity was defined as the amount of enzyme required to release 1 µmol of reducing sugar per minute. Protein determination was assayed according to the Bradford method [37]. 

### 3.5. Preparation of XynB2^Y509E^-CLEAs

4 mL of semi-purified enzyme was precipitated with ammonium sulfate to a saturation of 90%. After 30 min of stirring with a magnetic bar, glutaraldehyde (pH 6.5 of pH 8.5) was added slowly at a final concentration of 12.5 mM. The cross-linking reaction was maintained for 3 h at 25 °C and under 200 rpm stirring. The XynB2^Y509E^-CLEAs were recovered by centrifugation of the reaction mixture at 1840× *g* for 5 min at 4 °C. The CLEAs were washed three times with CFG buffer (0.1 M, pH 6.5) to remove the excess of glutaraldehyde and unbound enzyme. The supernatants and washing solutions were preserved for subsequent activity determination using standard protocols. The XynB2^Y509E^-CLEAs was resuspended in 4 mL of CFG buffer (0.1 M, pH 6.5) and stored at 4 °C. The percentage of recovered activity and aggregation protein yield [12] from CLEAs were calculated using Equations (1) and (2), as indicated below.
(1)$$\mathrm{Recovered}\;\mathrm{activity}\;\left(\%\right)=\frac{\mathrm{CLEAs}\;\mathrm{total}\;\mathrm{activity}}{\mathrm{free}\;\mathrm{enzyme}\;\mathrm{total}\;\mathrm{activity}}\times 100$$
(2)$$\mathrm{Aggregation}\;\mathrm{protein}\;\mathrm{yield}\;\left(\%\right)=\frac{\mathrm{the}\;\mathrm{protein}\;\mathrm{concentration}\;\mathrm{in}\;\mathrm{CLEAs}\;\left(\frac{\mathrm{mg}}{\mathrm{mL}}\right)}{\mathrm{Initial}\;\mathrm{protein}\;\mathrm{concentration}\;\left(\frac{\mathrm{mg}}{\mathrm{mL}}\right)}\times 100$$

### 3.6. Optimization of XynB2^Y509E^-CLEAs Preparation

In the precipitation step, several precipitating agents were tested. Acetone, butanol, and ammonium sulfate were added to XynB2^Y509E^ solution to give different final concentrations (70–90 *v*/*v*). After continuous stirring for 30, 60, and 120 min at 4 °C, the precipitates were recovered by centrifugation (1840× *g* for 5 min at 4 °C) and dissolved in a CFG buffer (0.1 M, pH 6.5). Different concentrations of glutaraldehyde as the crosslinking agent (12.5, 25, and 40 mM) were used. The effects of time for cross-linking efficiency were also determined for 2 and 4 h.

### 3.7. Scanning Electron Microscopy of XynB2^Y509E^-CLEAs

The scanning electron micrographs of CLEAs were recorded using a SEM HITACHI S-3500N (Hitachi, Tokyo, Japan). To prepare the sample, XynB2^Y509E^-CLEAs solution was spread over a microscope slide. The slide was dried at room temperature and then put inside a desiccation chamber to dry overnight. After drying, the sample was coated with gold using a BAL-TEC SCD 005 sputter coater (BAL-TEC GmbH, Schalksmühle, Germany).

### 3.8. Biochemical Characterization of XynB2^Y509E^-CLEAs

The biochemical characterization was carried out with the XynB2^Y509E^-CLEAs prepared at pH 8.5, measuring its ability to degrade p-PNX as substrate.

#### 3.8.1. Optimal Temperature and Thermal Stability of Free and XynB2^Y509E^-CLEAs

The optimal temperature for both free and immobilized enzymes were determined by assaying enzymatic activity in 0.1 M CFG buffer (pH 6.5) at different temperatures (40–80 °C) for 5 min on p-NPX 2.2 mM as substrate. To investigate the thermal stability, the free and immobilized β-xylosidase were incubated in 0.1 M CFG buffer (pH 6.5) at different temperatures (30–90 °C) for 60 min. After that, the residual enzymatic activity was measured by the method described above. 

#### 3.8.2. Effect of pH on Activity

The determination of the pH value for the maximum activity of the free and immobilized enzyme was determined by reaction with p-NPX substrate (2.2 mM) prepared in 0.1 M CFG buffer, which was adjusted in the range of pH 4.3 to 11. The pH stability of the enzymes was also investigated by measuring the residual activity after incubating the free and immobilized enzymes at 4 °C for 1 h at various pH values (pH 4.3–11.0) using a 0.1 M CFG buffer. All these reactions were performed according to the method described above. 

#### 3.8.3. Kinetic Parameters of Free and XynB2^Y509E^-CLEAs

The kinetic constants (*K*_m_ and *V*_max_ values) for the free and immobilized enzymes were determined by measuring the enzymatic activity in a 100 mM CFG buffer (pH 6.5) at 50 °C with different substrate concentrations (0.22–2.00 mM). *K_m_* and *V_max_* were calculated from the hyperbolic adjustment using the Origin 8.0 program (Originlab Corporation, Inc., Northhampton, MA, USA). All the results were carried out in triplicate.

#### 3.8.4. Recyclability of XynB2^Y509E^-CLEAs

The reusability of XynB2^Y509E^-CLEAs was assessed by measuring the β-xylosidase activity in ten consecutive experimental reaction cycles. The reaction using 5 µL of CLEAs and 2.0 mM of pNPX in 100 mM CFG buffer (pH 6.5) was carried out at 50 °C for 5 min. After that, CLEAs were recovered from the reaction mixture by centrifugation (at 2650× *g* for 3 min at 4 °C). The pellets were washed three times with CFG buffer (0.1 M pH 6.5) and then suspended again in a fresh reaction mixture to measure a new cycle of use. The activity determined on the first cycle was considered as control (100%) for the calculation of the remaining percentage of activity after repetitive uses.

## 4. Conclusions

In the present study, a recombinant mutant of the family 52 glycoside hydrolase was produced, kinetically characterized, and its CLEAs were then successfully prepared using 90% of ammonium sulfate as the precipitant and 12.5 mM glutaraldehyde for 3 h at room temperature and pH 6.5 or 8.5. It is remarkable that the XynB2^Y509E^-CLEAs prepared at pH 8.5 exhibits considerable promises as an immobilized biocatalyst, since it preserves both the xylanolytic activities that the Y509E mutant displayed as a free form. The XynB2^Y509E^-CLEAs synthetized displayed higher pH and thermal stability than the free enzyme. Moreover, the CLEAs also rendered good reusability, which was sustained by the fact that the CLEAs retain more than 60% of their original activity after re-use for ten batches. Finally, since the CLEAs prepared in this work were produced from a semi-crude β-xylosidase extract, we plan to carry out a comparison with CLEAs obtained using a purified enzyme. This future research is focusing on determining whether the catalytic efficiency of CLEAs will be higher than the free enzyme.

## Figures and Tables

**Figure 1 molecules-26-00451-f001:**
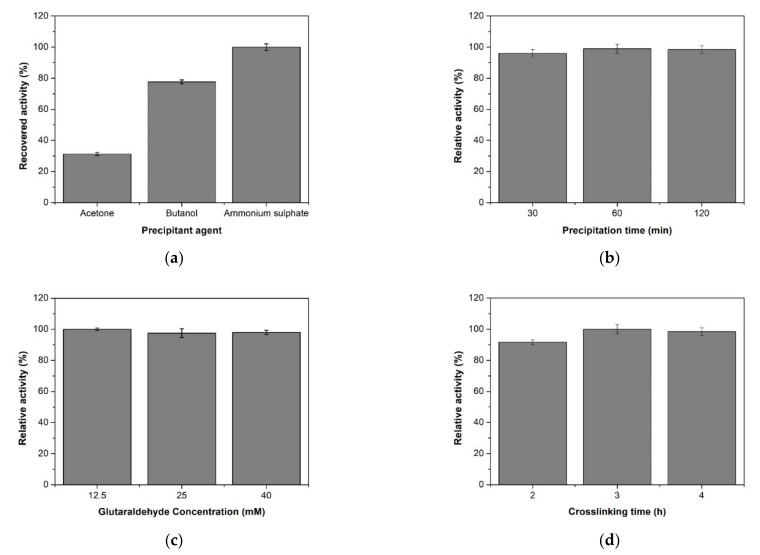
(**a**) Precipitation of XynB2^Y509E^ with different precipitant agents. (**b**) Precipitation of XynB2^Y509E^ at different times using 90% of ammonium sulfate. (**c**) Concentration of glutaraldehyde for cross-linking reaction. (**d**) Time of cross-linking using 12.5 mM of glutaraldehyde. All values are the mean of three replicates and error bars represent standard errors.

**Figure 2 molecules-26-00451-f002:**
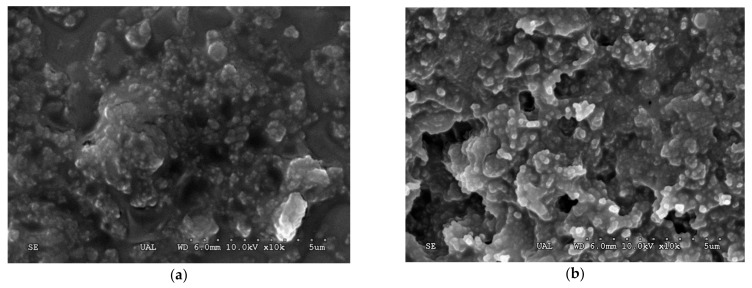
Scanning electron microscopy (SEM) images of cross-linked enzymatic aggregates (CLEAs) prepared at pH 6.5 (**a**) and pH 8.5 (**b**).

**Figure 3 molecules-26-00451-f003:**
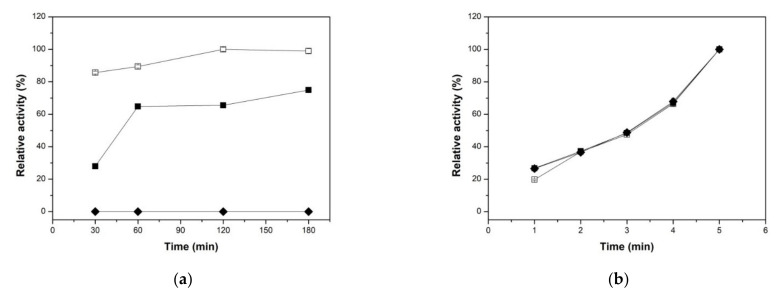
Effect of reaction time on the xylanase (**a**) and β-xylosidase (**b**) activities showed by soluble XynB2^Y509E^ (□) and immobilized XynB2^Y509E^-CLEAs prepared at pH 8.5(■) or pH 6.5 (♦). Substrate concentrations: Xylan (1%) or p-NPX (2.2 mM). The data points represent the mean values of three replicates with the SE indicated as error-bars. XynB2^Y509E^-CLEAs prepared at pH 6.5 or at pH 8.5 displayed the same β-xylosidase activity than that the soluble XynB2^Y509E^.

**Figure 4 molecules-26-00451-f004:**
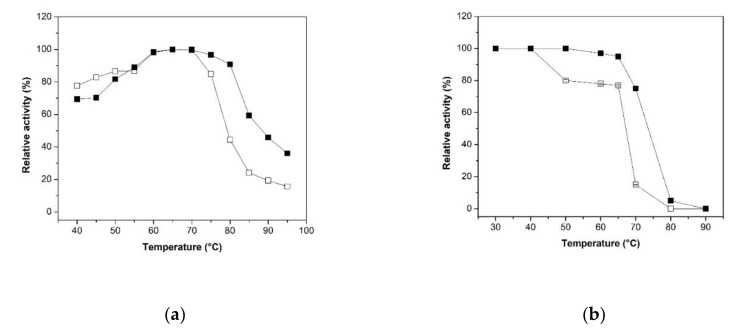
Effect of temperature on the β-xylosidase activity and stability of the free XynB2^Y509E^ and immobilized XynB2^Y509E^-CLEAs. (**a**) Effect of temperature on the activity of the free XynB2^Y509E^ (□) and immobilized XynB2^Y509E^-CLEAs (■) when the reactions were performed in the citrate phosphate-glycine (CFG) buffer (0.1 M, pH 6.5) using the chromogenic assay described in the experimental section. Relative activity was referred to the highest activity of the enzyme. (**b**) Effect of temperature on enzyme stability of the free XynB2^Y509E^ (□) and immobilized XynB2^Y509E^-CLEAs (■) when the reactions were performed at 50 °C after the enzymes were incubated for 1 h at indicated temperature. Relative activity was referred to the initial activity of the enzyme, prior to any incubation using the standard activity assay.

**Figure 5 molecules-26-00451-f005:**
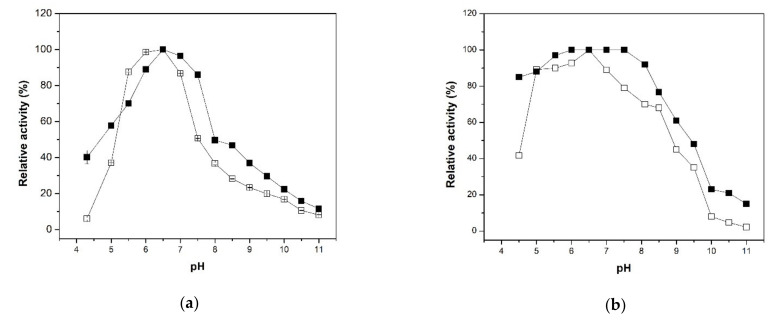
Effect of pH on the activity and stability of the free XynB2^Y509E^ and immobilized XynB2^Y509E^-CLEAs. (**a**) Effect of pH on β-xylosidase activity of the free XynB2^Y509E^ (⊞) and immobilized XynB2^Y509E^-CLEAs (■) when the reactions were performed in CFG buffer (0.1 M) and 50 °C using the chromogenic assay described in the experimental section. Relative activity was referred to the highest activity of the enzyme. (**b**) Effect of pH on enzyme stability of the free XynB2^Y509E^ (□) and immobilized XynB2^Y509E^-CLEAs (■) when the reactions were performed at 50 °C after the enzymes were incubated for 1 h at the indicated pH using as substrate p-NPX 2.2 mM, pH 6.5. Relative activity was referred to the initial activity of the enzyme, prior to any incubation using the standard activity assay.

**Figure 6 molecules-26-00451-f006:**
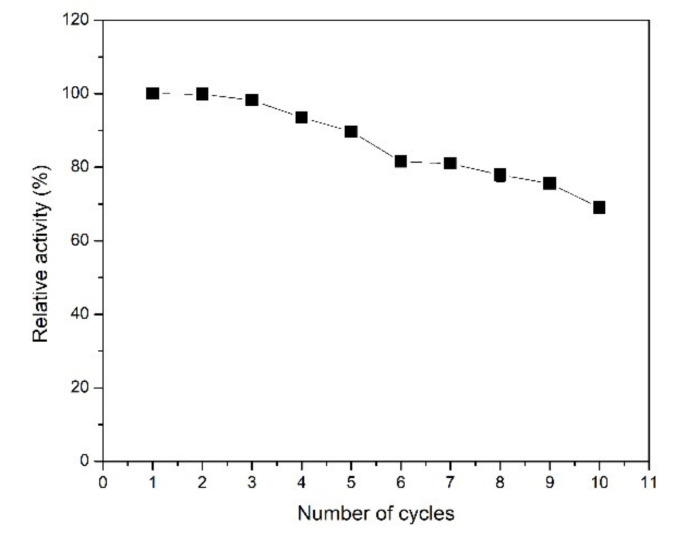
Reusability of XynB2^Y509E^-CLEAs.

**Table 1 molecules-26-00451-t001:** Kinetic parameters for the β-xylosidase activity of free XynB2^Y509E^ and XynB2^Y509E^-CLEAs.

Enzyme	*K_m_* (mM)	*V_max_* (µmol/min)
Free XynB2^Y509E^	0.9 ± 0.1	1.6 ± 0.2
XynB2^Y509E^-CLEAs	1.2 ± 0.2	1.18 ± 0.06

The activity was assayed in CFG buffer (100 mM, pH 6.5) at 50 °C.

## Data Availability

The data presented in this study are available on request from the corresponding author.

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
