# Peer review of "Characterization of Cross-Linked Enzyme Aggregates of the Y509E Mutant of a Glycoside Hydrolase Family 52 β-xylosidase from G. stearothermophilus"

_molecules, 2021, doi:10.3390/molecules26020451_

Round 1
Reviewer 1 Report
In this manuscript, the authors prepared CLEA with the Y509E mutant of glycoside hydrolase. The cross-linking conditions were optimized and 92.3% of the original β-xylosidase activity was obtained. Then the CLEA was characterized, including the effects of temperature, pH and reusability on the activity. The manuscript has some merits that deserved publication after solving the following issues:
1) References regarding CLEA can be updated with more recent references. For example, (Catalysts. 2018, 8: 460; Int J Biol Macromol. 2020, 144: 1013).
2) In the cross-linking process, the changes in protein structure will affect the enzyme’s activity. So I suggest the authors investigate the protein recovery to evaluate the process.
3) The authors claimed that the mutation introduced the exo-xylanase activity, but there was no explanation in the results and discussion although the experiments were performed. What effect of the mutation on the immobilization?
4) There are some spelling and grammar errors; check and correct them.
Author Response
In this manuscript, the authors prepared CLEA with the Y509E mutant of glycoside hydrolase. The cross-linking conditions were optimized and 92.3% of the original β-xylosidase activity was obtained. Then the CLEA was characterized, including the effects of temperature, pH and reusability on the activity. The manuscript has some merits that deserved publication after solving the following issues:
We thank the reviewer for finding our manuscript with merits to be published in Molecules. Your valuable time for evaluating complete manuscript as well as your comments and suggestions are highly appreciated, as they have helped us to improve the quality of our manuscript. In the revised manuscript, we have also made changes according to your comments trying at our best to provide the required explanation as well.
Point 1: References regarding CLEA can be updated with more recent references. For example, (Catalysts. 2018, 8: 460; Int J Biol Macromol. 2020, 144: 1013).
Response 1: As per the reviewer suggestion, we have updated the references regarding CLEAs. In this sense, we have added two sentences in the first paragraph of the introduction section of the revised version of our manuscript to indicate the importance of enzyme immobilization and to emphasize the robustness of CLEAs as a carrier-free immobilization technique. We have included a new reference which now is considered as the reference number 1.
[1] Xu, M.Q.; Wang, S.S.; Li, L.N.; Gao, J.; Zhang, Y.W. Combined cross-linked enzyme aggregates as biocatalysts. Catalysts. 2018, 8, 460. doi: 10.3390/catal8100460
We also added new sentences to the end of the third paragraph regarding the scope of CLEA methodology to form various catalyzing reactions with a single CLEA particle. In consequence, we have modified the manuscript including two new more references cited below.
[11]Periyasamy, K.; Santhalembi, L.; Mortha, G.; Aurousseau, M.; Subramanian, S. Carrier-free co-immobilization of xylanase, cellulase and β-1,3-glucanase as combined cross-linked enzyme aggregates (combi-CLEAs) for one-pot saccharification of sugarcane bagasse. RSC Adv. 2016, 6, 32849–32857. doi: 10.1039/C6RA00929H
[12] Xu, M.Q.; Li, F.L.; Yu, W.Q.; Li, R.F.; Zhang, Y.W. Combined cross-linked enzyme aggregates of glycerol dehydrogenase and NADH oxidase for high efficiency in situ NAD+ regeneration. Int J Biol Macromol. 2020, 144, 1013-1021. doi: 10.1016/j.ijbiomac.2019.09.178.
In our previous version, the reference 18 cited as Bhattacharya & Pletschke (5th paragraph of the Introduction section) was miscited in the reference list. We apologize for this. In the revised manuscript the reference was corrected.
As per suggestion about updating the references regarding CLEA, we also took advantage to include three new sentences to the fifth paragraph of the Introduction section. These sentences are regarding of a combi-CLEAs of xylanase and mannanase that was successfully applied in the conversion of lignocellulose biomass. This new information was now cited as reference 22 in the reference list.
[22] Bhattacharya, A.; Pletschke, B.I. Strategic optimization of xylanase–mannanase combi-CLEAs for synergistic and efficient hydrolysis of complex lignocellulosic substrates. J. Mol. Catal. B Enzym. 2015, 115, 140-150. doi: 10.1016/j.molcatb.2015.02.013
The numberings of the rest of the references have been updated
Point 2: In the cross-linking process, the changes in protein structure will affect the enzyme’s activity. So I suggest the authors investigate the protein recovery to evaluate the process.
Response 2: We agree with the reviewer in that during the cross-linking process, the changes in protein structure will affect the enzyme’s activity. Glutaraldehyde is the cross-linking agent most frequently used for the preparation of CLEA. However, with some enzymes, low or no activity recovery has been observed with glutaraldehyde cross-linking (Sheldon, 2019; Ref 5 in the previous version of the manuscript). Talekar et al. (Ref 3 in the previous version of the manuscript) have suggested that for the preparation of CLEAs enzymes, in which lysyl amino groups are essential for their activity and when the substrates are macromolecules, is better to use macromolecular cross linkers instead of glutaraldehyde. Given that the catalytic nucleophile and acid/base residues for the β-xylosidase activity of our enzyme are Glu335 and Asp495, respectively, and the fact that XynB2 significantly prefers small substrates with xylose at the glycone moiety (i.e. 4-nitrophenyl β-D-xylopyranoside) we decided to use glutaraldehyde as cross linking agent.
We had stated in our previous version of the manuscript about the CLEAs preparation results that (Page 3, lines 125-126) “Given that some enzymes are inactivated by glutaraldehyde, the cross-linker amount and cross-linking time on the activity of the resultant CLEAs must be optimized”. Therefore, we tested the optimum quantity of glutaraldehyde required for activity recovery by adding glutaraldehyde (GA) to final concentrations of 12.5, 25 and 40 mM, which correspond to 0.24%, 0.47%, and 0.75% (v/v) GA. These concentrations were selected taking into account the crosslinking concentrations of GA used for the preparation of other cross-linked xylanase aggregates previously reported. We found that these three different glutaraldehyde concentrations showed the maximum of the relative activity expected, as concluded from Figure 1C. This finding was indicated in page 3 of the previous version (lines 131-133, Results and discussion section).
We have followed the suggestion of the reviewer about investigating the protein recovery to evaluate the cross-linking process. To accomplish this, we determine the immobilization yield in terms of protein recovery using the equation described by Xu and co-workers in “Combined cross-linked enzyme aggregates of glycerol dehydrogenase and NADH for high efficiency in situ NAD+ regeneration” (2020) Int. J. Biol. Macromol., 144, 1013-1021 (https://doi.org/10.1016/j.ijbiomac.2019.09.178). This paper was cited as number 12 in the revised version of the manuscript. The aggregation yield value determined was found to be 87.3% for the XynB2Y509E-CLEAs prepared with 90% ammonium sulfate during 30 min and crosslinking using 12.5 mM glutaraldehyde at pH 8.5. Therefore, the last value allows us to conclude that the cross-linking process is effective. Furthermore, the new result presented is completely consistent with the high value of recovered activity in XynB2Y509-CLEAs. We have added several sentences on page 5 to describe and discuss this new finding (Results and discussion section, 2.1.2).
In addition, in the Methods section of the revised version, we have added and cited the aggregation protein yield equation as Eq. 2. We also have now written a sentence in the paragraph of “enzymatic assays” to indicate and cite the method that we used to calculate the protein concentration.
[24] Bradford, M.M. A rapid and sensitive method for the quantitation of microgram quantities of protein utilizing the principle of protein-dye binding. Anal. Biochem. 1976, 72, 248-254. doi: 10.1016/0003-2697(76)90527-3
The numberings of the rest of the references have been updated.
Point 3: The authors claimed that the mutation introduced the exo-xylanase activity, but there was no explanation in the results and discussion although the experiments were performed. What effect of the mutation on the immobilization?
Response 3: We appreciate the concern raised by the reviewer.
XynB2 from Geobacillus stearothermophilus T6 strain has been considered as a representative member of family 52 glycoside hydrolases (see for instances Ref 10-12 in the previous version of the manuscript or Ref 13-15 in the revised manuscript). It has been characterized as a β-xylosidase with high specificity for xylose as the glycone moiety. In 2014, Huang and co-workers cloned a xylosidase gene from another strain of G. stearothermophilus which encodes a β-xylosidase that shares 99.3% of identity with XynB2 (Ref 16 in the previous version of the manuscript or Ref 19 in the revised manuscript). In this paper, Huang and co-workers show that the mutation in the Y509 site of XynB2 into glutamic acid produce the enzyme variant, Y509E, displays not only xylosidase activity but also xylanase activity. Both β-xylosidase and xylanase activities displayed by the Y509E variant were extensively characterized in that paper as well.
As stated in lines 74-76 of our previous version of the manuscript, “CLEAs were prepared in order to evaluate the effect of the immobilization as CLEAs on the b-xylosidase ability of the mutant Y509E”. Thus, in the Materials and Methods regarding the biochemical characterization of XynB2Y509E-CLEAs we indicated that the enzymatic activity was assayed just using p-NPX as substrate (page 10, lines 354-381, Materials and Methods section). We apologize in advance if the enzymatic method to characterize our CLEAs on the first manuscript was poorly described. In the revised manuscript, we have now written a sentence in the general conclusion of paragraph in section 2.2.1 to indicate that the biochemical characterization of the CLEAs is related just to its β-xylosidase activity.
After adding the sentence in the section 2.2.1, we realized that the title of paragraph 2.2 on page 5 was incorrectly copied during manuscript upload. We regret this involuntary mistake. Thus, the correct title “Biochemical characterization of free XynB2Y509E and XynB2Y509E-CLEAs” has been now incorporated in the revised version of the manuscript.
From the data presented in Figure 3, we think that that the mutation performed on XynB2 does not have any effect on the process of immobilization carried out in our work. Our last statement is supported by the fact that XynB2Y509E-CLEAs prepared at pH 6.5 or pH 8.5 exhibited the same β-xylosidase activity than the soluble XynB2Y509E form (Figure 3b in the first version of the manuscript). However, we would like to pinpoint that the effect of the mutation on the immobilization was not analyzed in the present study. As first step, we found appropriate to limit the study only analyzing the mutant enzyme and comparing the CLEA with the free enzyme. In another step, once the mutant enzyme has been completely characterized, it is foreseen to perform further analysis immobilizing the native enzyme. How profound the effect of the mutation over immobilization is, still needs to be determined, therefore we agree with the reviewer’s question. However, given the current pandemic, the setup for these experiments has been postponed. We do hope to publish the expected results in an upcoming paper.
Point 4: There are some spelling and grammar errors; check and correct them.
Response 4: The manuscript has been checked for English, as suggested.
Reviewer 2 Report
The immobilization of a thermostable β-xylosidase as cross-linked enzyme aggregates is reported in the work. A recombinant mutant of the family 52 glycoside hydrolase was produced and kinetically characterized. Optimal conditions for its CLEAs have been successfully developed. The results obtained suggest that XynB2Y509E-CLEAs have a good operational stability, however future research is necessary in order to have a wider view of scope of its utility in industrial processes.
The authors made a careful analysis of the literature data and presented their original method of obtaining XynB2Y509E-CLEAs. In my opinion, the manuscript will be ready for publication after minor linguistic stylistics corrections.
Author Response
The immobilization of a thermostable β-xylosidase as cross-linked enzyme aggregates is reported in the work. A recombinant mutant of the family 52 glycoside hydrolase was produced and kinetically characterized. Optimal conditions for its CLEAs have been successfully developed. The results obtained suggest that XynB2Y509E-CLEAs have a good operational stability, however future research is necessary in order to have a wider view of scope of its utility in industrial processes.
The authors made a careful analysis of the literature data and presented their original method of obtaining XynB2Y509E-CLEAs. In my opinion, the manuscript will be ready for publication after minor linguistic stylistics corrections.
Response: We thank the reviewer for the positive comments on our manuscript. The manuscript was checked for English, as suggested.
Reviewer 3 Report
Summary: Cross-linked enzyme aggregates (CLEA) are valuable for industrial purposes where they convert lignin to ethanol and xylitol. Cross-linked enzymes have the advantage of stability, which makes them reusable. The present work manufactured a mutant enzyme that has both xylanase and xylosidase activity. Optimal conditions for preparing cross-linked enzyme aggregates were developed. Soluble free enzyme was compared to cross-linked enzyme aggregates in terms of activity, pH dependence, thermostability, and reusability. The cross-linked enzyme aggregates displayed higher pH and thermal stability and could be reused at least ten times.
Comments: The method developed in this manuscript is a valuable contribution, with potential to be useful for industrial applications.
Minor comments:
- Please describe citrate phosphate glycine buffer in terms of the grams of citric acid, grams of sodium phosphate, and grams of glycine in a stock solution of buffer.
- Page 9 line 316. What volume of 1 M sodium carbonate was added to stop the reaction?
- Page 9 Please give the extinction coefficient that was used to convert absorbance at 410 nm to units of xylosidase activity.
- Page 9 Please give the extinction coefficient for converting absorbance at 540 nm to units of xylanase activity.
- Figure 3b. It looks like pH 8.5 and pH 6.5 CLEAs as well as soluble enzyme all have the same xylosidase activity as a function of time. Please add a statement to the figure legend to make this point clear. The reader is unsure because comparison to Figure 3a shows that pH 8.5 CLEAs have a distinctly different activity from pH 6.5 CLEAs in the xylanase assay..
- Page 5 line 188 states that free XynB2 Y509E has an optimum reaction time of 30 min (Figure 3a). However, Figure 3a shows a relative activity of 85% at 30 min, and 100% at 120 min.
- Please add a statement to the methods section that CLEAs in Figures 4, 5, 6 and Table 1 were prepared at pH 8.5.
Author Response
Cross-linked enzyme aggregates (CLEA) are valuable for industrial purposes where they convert lignin to ethanol and xylitol. Cross-linked enzymes have the advantage of stability, which makes them reusable. The present work manufactured a mutant enzyme that has both xylanase and xylosidase activity. Optimal conditions for preparing cross-linked enzyme aggregates were developed. Soluble free enzyme was compared to cross-linked enzyme aggregates in terms of activity, pH dependence, thermostability, and reusability. The cross-linked enzyme aggregates displayed higher pH and thermal stability and could be reused at least ten times.
The method developed in this manuscript is a valuable contribution, with potential to be useful for industrial applications.
We thank the reviewer for the positive comments on our manuscript.
Minor comments:
Point 1: Please describe citrate phosphate glycine buffer in terms of the grams of citric acid, grams of sodium phosphate, and grams of glycine in a stock solution of buffer.
Response 1: In our work we used a 0.2 M stock solution of citrate phosphate glycine buffer system (58.82 g C6H5O7Na2.2H2O, 27.60 g NaH2PO4 and 15.02 g NH2CH2COOH in 1 L distilled water). p-NPX used to study the effect of pH on β-xylosidase activity was prepared in 0.1 M CFG buffer which was adjusted in the range of pH 4.3 to 11. We have added this information to Materials and Methods section.
Point 2: Page 9 line 316. What volume of 1 M sodium carbonate was added to stop the reaction?
Response 2: A volume of 100 µL of 1 M Na2CO3 was added to stop de reaction. We have added this information to Materials and Methods section.
Point 3: Page 9 Please give the extinction coefficient that was used to convert absorbance at 410 nm to units of xylosidase activity.
Response 3: A molar extinction coefficient of 18 mM-1 cm-1 for pNP at 410 nm was used. We have added this information to Materials and Methods section.
Point 4: Page 9 Please give the extinction coefficient for converting absorbance at 540 nm to units of xylanase activity.
Response 4: A molar extinction coefficient of 0.082 mM-1 cm-1 for xylose at 540 nm was used. We have added this information to Materials and Methods section.
Point 5: Figure 3b. It looks like pH 8.5 and pH 6.5 CLEAs as well as soluble enzyme all have the same xylosidase activity as a function of time. Please add a statement to the figure legend to make this point clear. The reader is unsure because comparison to Figure 3a shows that pH 8.5 CLEAs have a distinctly different activity from pH 6.5 CLEAs in the xylanase assay..
Response 5: As indicated by Figure 3b, the XynB2Y509E-CLEAs prepared at pH 6.5 or at pH 8.5 displayed the same activity than that the soluble XynB2Y509E. We have added a statement to the figure legend following reviewer´s suggestion.
In addition, we have added to the end of line 186 (page 5) of our previous version a sentence to indicate that the XynB2Y509E-CLEAs prepared at pH 6.5 did not display xylanase activity at any of the reaction time assayed.
Point 6: Page 5 line 188 states that free XynB2 Y509E has an optimum reaction time of 30 min (Figure 3a). However, Figure 3a shows a relative activity of 85% at 30 min, and 100% at 120 min.
Response 6: Reviewer is right, the optimum reaction time for the xylanase activity exhibited for free XynB2Y509E was 120 min. We have corrected the value of the optimum reaction time, and we thank the reviewer for noticing our mistake.
We stated at line 187 of our previous version of the manuscript that “The xylanase activity of the XynB2Y509E-CLEAs showed optimum reaction time of 60 min”. However, this statement is incorrect because as presented in Figure 3a the optimum reaction time for the xylanase activity of the XynB2Y509E-CLEAs is 180 min and not 60 min as stated. This inadvertent mistake was corrected in the revised version of the manuscript and we also added the phrase “prepared at pH 8.5” after the word XynB2Y509E-CLEAs.
Point 7: Please add a statement to the methods section that CLEAs in Figures 4, 5, 6 and Table 1 were prepared at pH 8.5.
Response 7: We agree with the reviewer´s suggestion to add a statement to the methods section that the CLEAs in Figures 4, 5, 6 and Table 1 were prepared at pH 8.5. Materials and Methods section of the previous version of the manuscript just included two subsections 3.1. for Materials and 3.2. for Methods. All the procedures and methodology used in our work was included into the same subsection 3.2. However, to accomplish the reviewer suggestion we have now considered each paragraph as a new subsection. Therefore, the four paragraphs between the lines 354 – 381 was include in a new subsection called “Biochemical characterization of XynB2Y509E-CLEAs”. There, we have added the statement suggested by the reviewer.
Round 2
Reviewer 1 Report
It could be accepted in the present form.